# MCA-YOLOV5-Light: A Faster, Stronger and Lighter Algorithm for Helmet-Wearing Detection

Cheng Sun [1,2,†], Shiwen Zhang [3], Peiqi Qu [3], Xingjin Wu [3], Peng Feng [4], Zhanya Tao [5], Jin Zhang [2,3,4,5,†] and Ying Wang [6,*]

1  School of Mathematics and Statistics, Hunan Normal University, Changsha 410081, China
2  Key Laboratory of Computing and Stochastic Mathematics (Ministry of Education), Hunan Normal University, Changsha 410081, China
3  College of Information Science and Engineering, Hunan Normal University, Changsha 410081, China
4  School of Computer and Communication Engineering, Changsha University of Science & Technology, Changsha 410114, China
5  Hunan Leran Intelligent Technology Ltd., Changsha 410221, China
6  School of Humanities and Management, Hunan University of Chinese Medicine, Changsha 410208, China
*  Correspondence: wy101223@163.com
†  These authors contributed equally to this work.

**Featured Application: Authors are encouraged to provide a concise description of the specific application or a potential application of the work. This section is not mandatory.**

**Abstract:** It is an essential measure for workers to wear safety helmets when entering the construction site to prevent head injuries caused by object collision and falling. This paper proposes a lightweight algorithm for helmet-wearing detection based on YOLOV5, which is faster and more robust for helmet detection in natural construction scenarios. In this paper, the MCA attention mechanism is embedded in the backbone network to help the network extract more productive information, reduce the missed detection rate of small helmet objects and improve detection accuracy. In order to ensure the safety of workers in construction, it is necessary to detect whether the construction workers are wearing safety helmets in real-time to achieve monitoring on-site. A channel pruning strategy is proposed on the MCA-YOLOv5 algorithm to compress it, realizing the optimal large-scale model into ultrasmall models for real-time detection on embedded or mobile devices. The experimental results on the public data set show that the model parameter volume is reduced by 87.2%, and the detection speed is increased by 53.5%, even though the MCA-YOLOv5-light reduces the mAP slightly.

**Keywords:** helmet-wearing detection; YOLOv5; attention mechanism; model pruning

## 1. Introduction

The construction industry is one of the most dangerous industries, and tragedies of not wearing the corresponding safety protective equipment (safety helmet, etc.) occur every year. According to the statistics from Eurostat and IBS in 2020, the number of non-fatal and fatal accidents in the construction industry always exceeds that in the manufacturing industry, especially head injury accidents at construction sites. Even though the number of head injuries is only about 7% of non-fatal accidents, they account for over 30% of fatal accidents, making it a significant issue and critical to the safety of construction workers [1]. The most common head injury on construction sites is traumatic brain injury (TBI), which can be fatal and which occurs mainly by the rapid acceleration or deceleration of the brain moving and colliding with the skull. According to the relevant investigation and research [2], the most common cause of TBI on construction sites is falling or being hit by objects.

In China, workers must wear safety helmets before entering each construction site. Therefore, workers must check that they are wearing their safety helmets. At present, the

daily helmet checking mainly relies on people, and the construction leaders stay at the construction site the whole day, which no doubt wastes a lot of human resources. Meanwhile, the proportion of construction site workers is severely imbalanced. Only relying on human supervision to ensure the safety of the construction personnel is impossible. In 2019, according to Sharma et al. [3], a survey showed that 80 percent of workers believed work-related traumatic brain injury (TBI) can be prevented and that 50 percent said they did not receive safety training, while more than half had no safety care at work. Brolin et al. [4] show that construction workers wearing safety helmets at work can reduce the impact of falling accidents and relieve traumatic brain injuries. Recognizing that head injury is an important factor affecting the safety of construction sites, legal regulations on the use of personal protective equipment (PPE) require employers to provide personal protection to employees. However, the workers themselves have a limited understanding of the importance of personal protective equipment and lack safety awareness, obviously, so it is essential to strengthen the care of the construction personnel.

At the present stage, the target detection algorithm can be divided into two mainstream directions: one is the one-stage detection algorithm based on the regression strategy, such as the RetinaNet algorithm [5], YOLO series algorithm [6–10] and the SSD algorithm [11] class. The core theory of such algorithms is to input the image into the model and directly return to the target's boundary target anchor box, position, and category information. The other is to generate regional recommendations, mainly in the R-CNN series [12–14], Libra R-CNN algorithm [15] and the Grid R-CNN algorithm [16] class. Such algorithms mainly generate a region of the candidate region from the image at the first stage and then generate the final target boundary anchor box from the candidate region at the second stage. At present, the research of algorithms in the field of deep learning is progressing rapidly, and more and more researchers want to combine deep learning algorithms with practical applications. For instance, Rao [17] combined the SAS module of the channel attention mechanism with the lightweight algorithm YOLOv3-tiny to obtain more helmet information features, improve the algorithm detection performance, and reduce the number of parameters and computation in the model. Yan et al. [18] replaced the traditional convolution algorithm in Darknet-53 with deep separable convolution to reduce feature loss and adjust model parameters, and, secondly, add the multiscale feature- fusion structure to obtain more shallow information, thus improving the accuracy of helmet-wearing detection. Fan et al. [19] added a convolutional layer to the Faster R-CNN network to detect targets with smaller pixel values. Mozaffari [20] proposed IrisNet, used to segment and detect objects. More and more researchers focus on object detection in small object detection. Y. Lee et al. [21] proposed a VoVNet based on improved DenseNet to improve the efficiency of object detection while retaining the benefits of cascaded aggregation of object detection tasks, improving the accuracy of object detection at smaller FLOPS. Stojnić V et al. [22] proposed using synthetic video to train CNN, effectively and accurately detect flying bees and improve the F1 score to 86%. Guimei Cao et al. [23] introduced contextual information into SSD to detect a small object. In addition, there are also some olfactory neural networks [24,25], etc. for detection.

Although the optimizations of the algorithm are helpful to improve the average accuracy of algorithm detection, there are still a series of disadvantages: the works of literature above lack the consideration of complexity, so the robustness of the improved algorithms is poor, especially in [17]. Even though the original ordinary convolution is replaced by a deep-separation convolution and an SAS module of channel attention mechanism, the algorithm structure is slightly redundant, and the required hardware configuration is higher, which is not useful to the real-time supervision and the application of construction scenarios. Meanwhile, considering the requirements of monitoring real-time performance, ref. [18] adds a multiscale feature fusion structure and [19] adds a convolution layer. There is a redundant structure in the algorithm, so there is still room for improvement in the real-time detection rate of the above algorithms.

General object detection cannot fit a specific scene well, so the corresponding detection task cannot be completed well. Common methods of adding attention mechanisms, in addition to adding additional computational overhead, cannot better connect contextual information. Due to the limited computational overhead, we adopt the multispectral channel attention (MCA) strategy, use the discrete cosine transform (DCT) in the channel attention mechanism to compress the channel and use the model-pruning method for sparse training to achieve higher detection accuracy with smaller models in the hat-wearing detection task. The reasons are as follows: (1) The global average pooling operation cannot express rich feature information [26]. GAP is a special case of DCT, and it is embedded in the MCA framework, adding more frequency component information obtained from redundant channels, which can extract more useful information; (2) The CNN model has the problems of model size, running time occupying memory and large amount of calculation. After the model introduces the scaling factor regular term, many scaling factor regular terms of the obtained model will tend to 0. After sorting the absolute value of the scaling factor, subtracting the channel corresponding to the pruning percentage, then finetuning and repeating the network thinning, we get a compact small network.

In summary, to solve the problems existing in the current safety helmet-wearing detection algorithm, the main contributions of this article are as follows: (1) Aiming for the problem that the YOLOV5 cannot better detect the shielding safety helmet and the smaller safety helmet in the task, the MCA-YOLOV5 model of multispectral channel attention (MCA) module in feature extraction network is proposed; (2) For the large model unable to embed the mobile terminal problems, first, the model introduces L1 regularization constraints associated to the scaling factor in the BN layer of the model and performs the sparsity training. The channels are then sorted by importance, and the less important channels will then perform pruning operations. Finally, the trimmed model is fine-tuned and trained again. Through model pruning, the MCA-YOLOV5-Light safety helmet-wearing detection algorithm can not only reduce the hardware cost and model parameter scale but also perform rapid detection.

## 2. Related Work

### 2.1. Attention Machine

In the field of deep learning, methods to shift attention to the most important areas of the image and ignore irrelevant parts are called the attention mechanism [27]. In essence, using the human visual system can explain the intuition behind attention well. The attention mechanism mimics the unique brain-signal-processing system in the human visual system, just as the human eye usually turns its attention to the target area of their interest and tries its best to capture the detailed characteristics of the target in the area, thus ignoring much irrelevant information. It is the survival mode formed by long-term natural selection of human beings, which enables humans to quickly obtain useful information resources from massive information with limited attention. The human attention mechanism effectively improves the accuracy and efficiency of information resource processing and acquisition to a large extent.

The human visual system uses attention to analyze and recognize complex real-life situations more effectively. With the development of technology, more researchers use attention mechanisms to optimize deep learning techniques and improve the characteristics of deep learning patterns. In computer vision systems, attention mechanisms are implemented by adaptive weighted features based on the importance of their respective inputs and provide benefits in many visual tasks, such as pose estimation, super-resolution and 3D vision, as well as multimodal tasks.

Hu et al. [28] propose the SENet, in which attention is used to collect global information, capture channel relationships and improve representation. It adopts weight-calibrated feature fusion and cannot fully utilize the global context. Yang et al. [29] proposed the Ggated channel conversion (GCT) to collect information effectively and explicitly model channel relationships. Wang et al. [30] proposed the Efficient Channel Attention Module

(ECA), which used a one-dimensional convolution to determine the interactions between the channels rather than the dimensionality reduction. Qin et al. [31] rethought the global information captured from a compression perspective and analyzed the global averaging pool in the frequency domain. They showed that the global mean pool was a special case of a discrete cosine transform (DCT) and used this observation to propose multispectral attention channel attention. Guo [32] proposed a small-memory external attention based on two externals that was easy to imitate and share and could be easily realized with the help of only two bridged normalization layers and linear layers. The algorithm subverted the existing self-attention mechanism very quickly. Yang et al. [33] proposed SimAM (a simple, parametric-free attention module that directly estimated 3D weights rather than extending 1D or 2D weights), which also emphasized the importance of learning changing attention weights in learning channels and spatial domains. Misra et al. [34] were motivated by the spatial attention mechanism and proposed a triple attention mechanism, a lightweight but effective attention mechanism, which was mainly used to capture cross-domain interactions. Hou et al. [35] proposed a coordinate attention mechanism incorporating positional features in the channel attention mechanism, thus enabling the network to focus on large important regions at less computational cost.

### 2.2. Model Puring

Model pruning is cutting the pretrained network model to delete the "unimportant" neurons or connections in the network model according to a series of parameter evaluation criteria to reduce as much as possible the model's parameter size of the model without losing the model accuracy. According to the different model pruning objects, model pruning can be further refined into structured pruning and non-structured pruning [36]. Structured pruning removes entire neurons, filters or channels and tradeoff tailoring grouping parameters to facilitate better use of hardware and software optimized for intensive computation. Unstructured pruning is mainly about pruning a single parameter, producing a relatively sparse neural network. Although the parameters are small, the arrangement may not be conducive to accelerating the use of modern libraries and hardware. Model pruning includes structured pruning and unstructured pruning.

#### 2.2.1. Structured Pruning

Structured pruning achieves simultaneous neural network compression and acceleration by directly deleting the structured convolutional neural network part and is well supported by various off-the-shelf deep learning libraries. At present, the main mainstream methods of structured pruning are the following four [37]: channel pruning, filter-level pruning, layer pruning, and connection pruning. A schematic representation of the model pruning section is shown in Figure 1. Channel pruning mainly reduces the number of redundant channels in the middle layer of the deep neural network model, aiming to optimize the computing and storage requirements of the model; filter-level pruning mainly removes the unimportant filters in the deep neural network model; layer pruning refers to the removal of some selected layers from the network model to reduce the volume of the deep neural network model. In addition, existing research attempts to delete the deep neural network model to reduce the parameter size of the network model.

Unlike unstructured pruning, structured pruning is more of a remodeling process, intending to prune the whole block completely. To avoid zero input, only blocks with residual connections can be trimmed. Despite the limitations, block pruning can effectively eliminate deep redundancy in some architectures. Furthermore, it can be achieved in combination with filter pruning, resulting in higher pruning rates. In structured pruning, filter-level pruning outperforms unstructured pruning; however, unstructured pruning can achieve the optimal training speed.

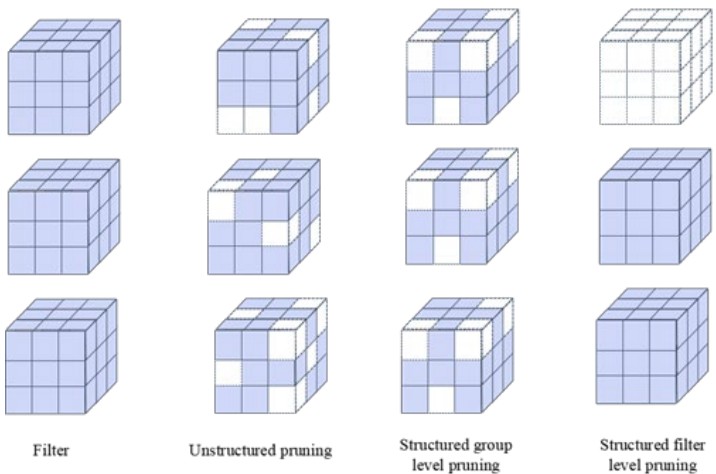

Filter    Unstructured pruning    Structured group level pruning    Structured filter level pruning

**Figure 1.** Schematic representation of model pruning.

### 2.2.2. Unstructured Pruning

Since the number of model parameters was not very large in earlier times, deep neural network model pruning work focused on convolution weights with zero unnecessary connections occupying most of the computations needed during execution. To ensure the consistency of the model structure, the weights can only go to zero and cannot be pruned. Therefore, weight pruning requires assigning specific coordinates for each weight, which is difficult to satisfy in today's trillion-level models. In addition, special hardware equipment is needed to speed up the training process. Early work on weight pruning [38] proposed a significant measure to remove redundant weights determined by the second derivative matrix of the loss function. Many methods are also proposed to determine weight-zero criteria, such as iterative threshold selection [39] and Hoffman code [40]. Kwon et al. [41] proposed a sparsely quantized neural network weight representation scheme, specifically implemented by fine-grained and unstructured pruning methods.

## 3. Method

### 3.1. MCA-YOLOv5

#### 3.1.1. MCA-YOLOv5 Safety Helmet Wearing Detection Algorithm

YOLOv5 is a single-stage target detection algorithm proposed in 2020 by Jocher et al. Due to the differences in model depth and width, YOLOv5 can be divided into four different versions: YOLOv5s, YOLOv5m, YOLOv5l and YOLOv5x. Moreover, the YOLOv5s network has the fastest computation speed but the lowest mean average accuracy, whereas the YOLOv5x network has the opposite characteristics.

The YOLOv5 network structure is divided into four parts: input end, backbone network, neck, and prediction part. YOLOv5 adds new mosaic data enhancement in the data input part; the Focus architecture and CSP architecture is mainly used in the backbone network; FPN + PAN architecture is added in the neck; replacing CIoU loss function with GIoU loss function, the loss function of boundary target anchor box is improved and in object detection, YOLOv5 detects multiple object anchor boxes by a weighted NMS algorithm. Therefore, YOLOV5 is suitable for multiscale object detection, and this paper optimizes the algorithm based on YOLOV5, making it suitable for safety helmet detection in construction site scenarios.

In the task of site scene helmet detection, due to the complex environment of the site scene, the situation of a worker wearing the helmet while it is not entirely blocked and the helmet being small relative to the background and therefore not easily detected occurs frequently. Based on the characteristics of the site scene, the MCA-YOLOV5 model is designed to improve the accuracy of the site helmet-wearing detection.

In the small object detection task, the collected small target feature information is also gradually weakened as the number of network layers increases gradually, so it is

easy to cause the algorithm to err and miss the detection of small targets. However, the MCA module uses the global average pooling and the remaining frequency components to enhance the features in the feature map so that the network strengthens the overlooked point learning of the target features during the training process. In this paper, the MCA module is embedded in the backbone network to enrich the network acquisition of features. Figure 2 shows the YOLOV5 structure diagram of the MCA module added.

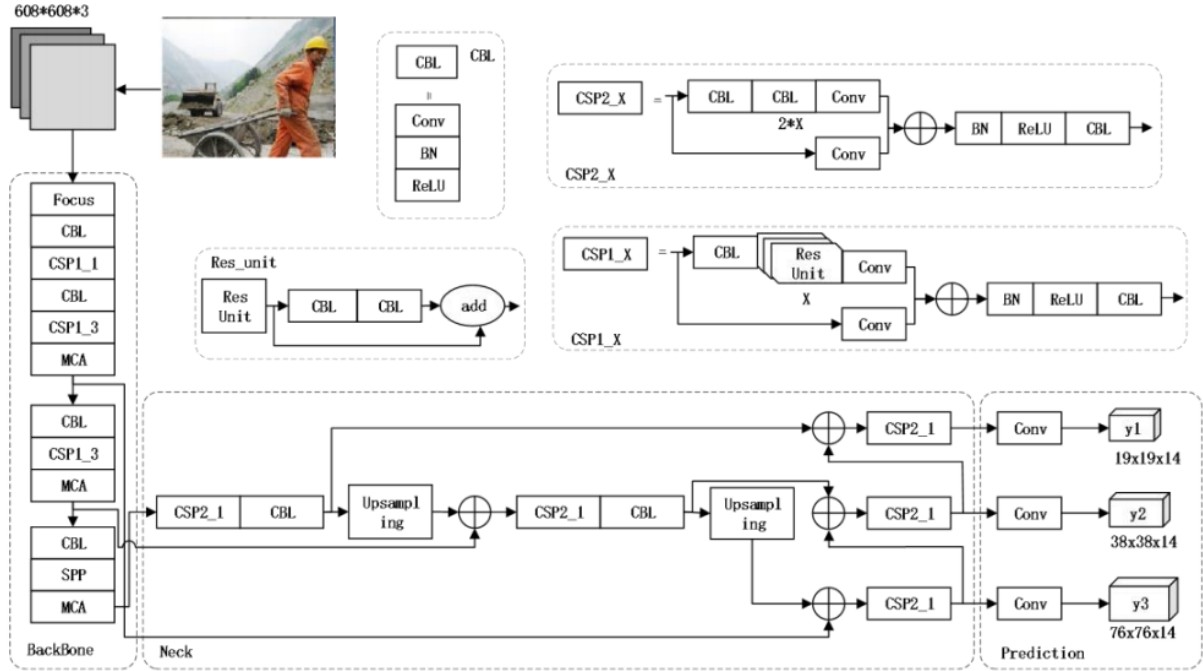

**Figure 2.** The MCA-YOLOv5 network model structure.

### 3.1.2. MCA Attention Mechanism

Due to the limited computational overhead, the weight function requires a scalar for each channel, and the Global Average Pooling (GAP) becomes a standard choice in the deep learning field due to its simplicity and efficiency. Although this choice is simple and efficient, the GAP cannot capture the diverse pattern information very well, so the feature generalization is missing in response to different inputs. Qin et al. [31] proposed the Multispectral Channel Attention (MCA) module to solve the above problems.

GAP is the lowest frequency of Discrete Cosine Transform (DCT), and only taking GAP is equivalent to not using the remaining frequency components containing much available information in the feature channel. 2DDCT is shown in Equation (1):

$$f_{h,w}^{2d} = 2DDCT(x_i) = \sum_{i=0}^{H-1} \cos\left(\frac{(2i+1)\pi h}{2H}\right) \left\{\sum_{j=0}^{W-1} x_{i,j}^{2d} \cos\left(\frac{(2j+1)\pi w}{2W}\right)\right\}$$
$$i, h \in \{0, 1, \ldots, H-1\}, \quad j, w \in \{0, 1, \ldots, W-1\} \tag{1}$$

in which $x \in \mathbb{R}^{H \times W}$ is the input and $H$ and $W$ represent the height and width of the input components.

Although the multispectral channel attention module has the same starting point as other attention modules, the multispectral channel attention module not only retains the global average pool but also uses the frequency component besides the global average pooling, which can solve the problem of missing information caused by only a single frequency, make the network model pay more attention to important features and filter redundant features. The multispectral channel attention module is improved based on the SE (Squeeze and Excitation) module. The specific structure diagram is shown in Figure 3.

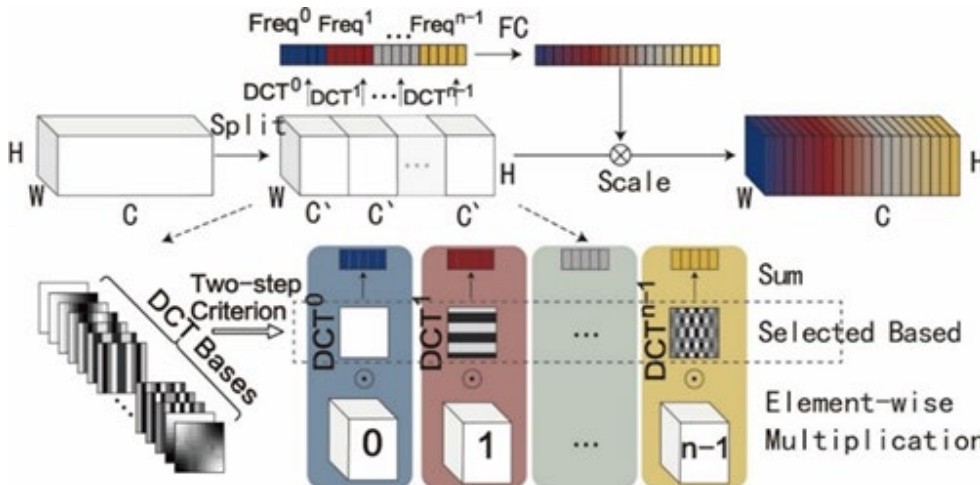

**Figure 3.** MCA Network Structure (The first step is to calculate the results of each frequency component in the channel attention separately; then, the frequency components are combined and the last step is to screen out the Top-k frequency components with the best performance according to the obtained results).

In order to integrate more frequency component information, including the lowest frequency component GAP, the multispectral channel attention module uses two-dimensional discrete cosine variables to operate. As can be seen from Figure 3, the specific execution steps are as follows: First, the input $X$ is divided into $n$ parts according to the channel dimension, which should be able to be divided by the number of channels. For each part, its corresponding two-dimensional discrete cosine variable frequency component is uniformly assigned, and its final output can be used as a preprocessing result of channel attention (e.g., GAP), as shown in Equation (2):

$$Freq^i = 2DDCT^{u,v}\left(X^i\right) = \sum_{h=0}^{H-1}\sum_{w=0}^{W-1} X^i_{:,h,w}B^{u,v}_{h,w} \qquad (2)$$

$Freq \epsilon R^C$ is the preprocessed multispectral vector, $2DDCT$ expressed as the 2-*DDCT* frequency component, $[u, v]$ is corresponding to the frequency component 2D index, $H$ is the height of input $X$, $W$ is the width of input $X$, $X^1, X^2, \ldots, X^{n-1}$ represents the divided parts, $X^i \in R^{C' \times H \times W}$, $C' = \frac{C}{n}$, $C$ can be divisible by $n$ and $B^{u,v}_{h,w}$ is the basis function of the 2D discrete cosine transformation.

Then, the frequency components of each part are combined, which is the multispectral vector, as shown in Equation (3): $Freq \in R^C$

$$Freq = cat\left(\left[Freq^0, Freq^1, \ldots, Freq^{n-1}\right]\right) \qquad (3)$$

Among them: $Freq \in R^C$ is the preprocessed multispectral vector.

$$att = sigmoid(fc(Freq)) \qquad (4)$$

*sigmoid* representing the sigmoid function, *fc* representing the mapping function, such as a fully connected layer or a one-dimensional convolution. *att* represents the entire MCA attention mechanism.

### 3.2. MCA-YOLOv5-Light Safety Hat Wearing Detection Model

Although the MCA-YOLOv5 algorithm can better test whether the construction workers in natural construction sites wear safety helmets, it has many detection model parameters and a large model volume, so it is difficult to apply in order to detect the construction

workers in mobile scenarios directly. In view of the detection problems of the above algorithm, we pruned the MCA-YOLOV5 algorithm and reduced the volume size and calculation amount of the above algorithm as much as possible without reducing the algorithm detection performance. That is, while ensuring the detection rate, the pruning algorithm should be able to receive higher-definition images for detection, which is also conducive to the application of mobile scenarios.

### 3.2.1. Sparsity Training of the Safety-Helmet-Wearing Detection Model

The training granularity of the sparse model training is divided into weight level, channel level and network level, as shown in Figure 4. The sparse training of the fine-grain level (e. g., weight level) has the highest compression rate and model flexibility, but it usually requires specialized hardware or underlying library to accelerate the inference model; the sparse network level does not require special hardware or underlying library for model acceleration but has significantly poor compressibility and flexibility for the model itself. Moreover, the sparse training of the network layer is fully effective only when the depth of the network model exceeds 50 layers [42]. In contrast, sparse training at the channel level achieves a good balance between flexibility and ease of implementation, which can be applied to any convolutional neural network or a fully connected network. For the above reasons, the pruning algorithm in this paper will sparsely train the model at the channel level.

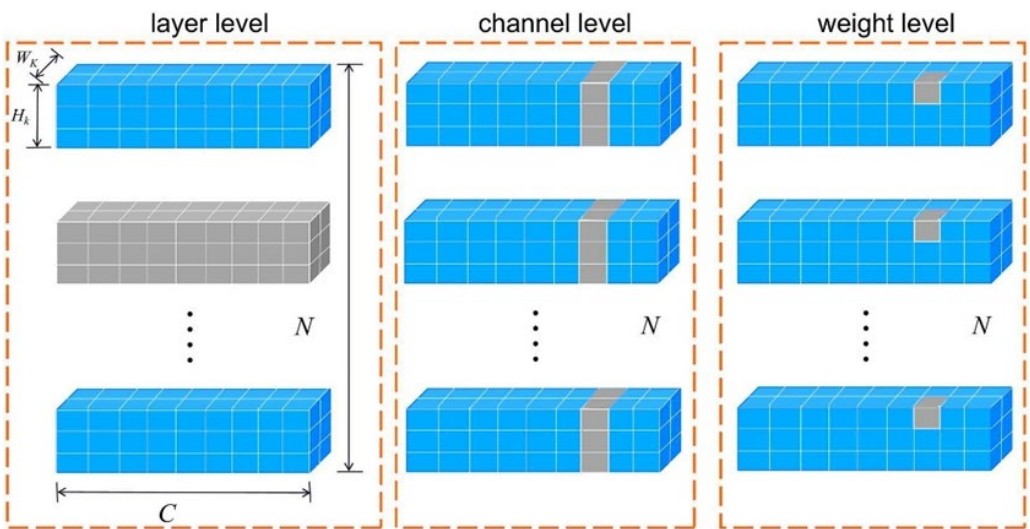

**Figure 4.** Visualization of different types of pruning (gray part represents pruning size).

This paper performs channel-level sparsity on the MCA-YOLOv5 helmet-wearing detection algorithm obtained from regular training. During sparsification, a simple L1 regularization term applied to the channel scaling factor of the BN is chosen to be sparse on the channel granularity of the model, resulting in good compression ratios without requiring specialized hardware or the underlying library. For the sparsity training, the scaling factor and the algorithm weights are trained simultaneously in this paper, and then the scaling factor is applied to the sparse regularization. When the sparse training is completed, the convolution channel with a slightly lower scale factor in the algorithm will be deleted, and the deleted model will be fine-tuned and trained to restore the accuracy of the model in the end. The objective function of the sparse training is specifically shown in Equation (5):

$$L = \sum_{(x,y)} l(f(x,W),y) + \lambda \sum_{Y \in r} g(Y) \tag{5}$$

Formula: $W$ is the weight of the trainable model, $(x, y)$ is the input and output of the model training. The first addition is the loss of the CNN network during normal training, $\lambda$ is the adjustment coefficient, and $g(\gamma)$ is the sparse penalty on the scaling factor. This chapter selects the L1 parametric as the penalty term, as shown in Equation (6):

$$g(Y) = |Y| \tag{6}$$

Batch normalization is a typical improvement measure in CNN networks and is usually placed before the activation layer can not only strengthen the network generalization performance but also accelerate the rate of network convergence. The formula is shown in Equation (7):

$$\hat{z} = \frac{z_{in}\mu_B}{\sqrt{\sigma_B^2 + \varepsilon}}; z_{out} = \gamma\hat{z} + \beta \tag{7}$$

where: $B$ represents the current minimum number of batches and $\mu_B$ and $\delta_B$ are the mean and standard deviation of $B$, and $\gamma$ and $\beta$ are the trainable affine transformation parameters (scale and displacement) and $z_{in}$ and $z_{out}$ are the input and output of the BN layer.

### 3.2.2. Channel Pruning and Fine-Tuning of the Safety-Helmet-Wearing Detection Model

At the 2017 ICCV Conference in the Field of Computer Vision, Liu et al. [21] presented a paper proposing a pruning algorithm for pruning convolutional neural networks in a simple but rather efficient manner. For the VGGNet network model, the network-slimming and pruning algorithm reduced its model size by 20 times and the computational operation by 5 times, whereas the pruned network model did not significantly reduce the accuracy. Drawing on the idea of this network-slimming pruning algorithm, this paper designs a pruning algorithm for a lightweight object detection network, which has pruned the convolutional channels in the detection network and can be directly used for object detection networks based on the convolutional neural network. At the same time, the pruned resulting model does not require the use of specialized hardware or the underlying library and can be directly used for rapid detection tasks.

The detailed process of the pruning algorithm proposed in this paper is shown in Algorithm 1 and Figure 5. It uses the scaling factor of the BN layer as the measure of model pruning and the convolution channel below the preset threshold to reduce the volume size of the algorithm by the pruning operation, then train the trimming algorithm to recover the accuracy and finally obtain a lightweight detection network. Figure 5 shows the specific process of the channel pruning algorithm. After the model is sparsely trained, the lower convolution channel (yellow in the figure) will be pruned. Because the network structure of the pruning model changes compared with the original model, whereas the neural network parameters learned by the original network structure do not change, the target detection power of the pruning model is reduced, and the mapping power of the pruning model is low. The pruned model is trained with fine-tuning when the model can relearn the neural network parameters according to the current network structure, thus recovering the model's detection accuracy and improving the mapping.

In conclusion, the MCA-YOLOv5-Light safety-helmet-wearing detection model mainly includes three stages: sparse training, model pruning, and model recovery. The entire pruning process is shown in Figure 6. In this paper, the MCA-YOLO v5 model is first sparsely trained, then the channel pruning operation is performed on the network model according to the pruning strategy and finally fine-tuning training is performed to obtain the final lightweight compression model.

---

**Algorithm 1** A channel pruning algorithm based on the scaling factor

---

     **Input:** Original model
**Output:** Lightweight compression model
1 bn_weights = $\gamma_1, \gamma_2, \gamma_3, \ldots \ldots \gamma_n$, Initialize Epochs = 200
2 //Turn on the sparse training session
3 while epoch $\leq$ Epochs
4      $g(Y) \leftarrow |Y|$
5      objective function $L \leftarrow \sum\limits_{(x,y)} l(f(x,W),y) + \lambda \sum\limits_{Y \in r} g(Y)$
6      Output of the BN $z_{out} \leftarrow Y\hat{z} + \beta$
7      Update $\gamma_1, \gamma_2, \gamma_3, \ldots \ldots \gamma_n$
8      bn_weights$\leftarrow \gamma_1, \gamma_2, \gamma_3, \ldots \ldots \gamma_n$
**9 end**
10 Initialize prune_ratio = 0.9
11 To $\gamma_1, \gamma_2, \gamma_3, \ldots \ldots \gamma_n$ Sort by channel number
12 if $c_1, c_2, c_3, \ldots \ldots, c_n$ channel_id < len(channels) prune_ratio$\times$
13      Trim the channel $c_1, c_2, c_3, \ldots \ldots, c_i$
**14 end**
15 Fine-tuning of the pruning network

---

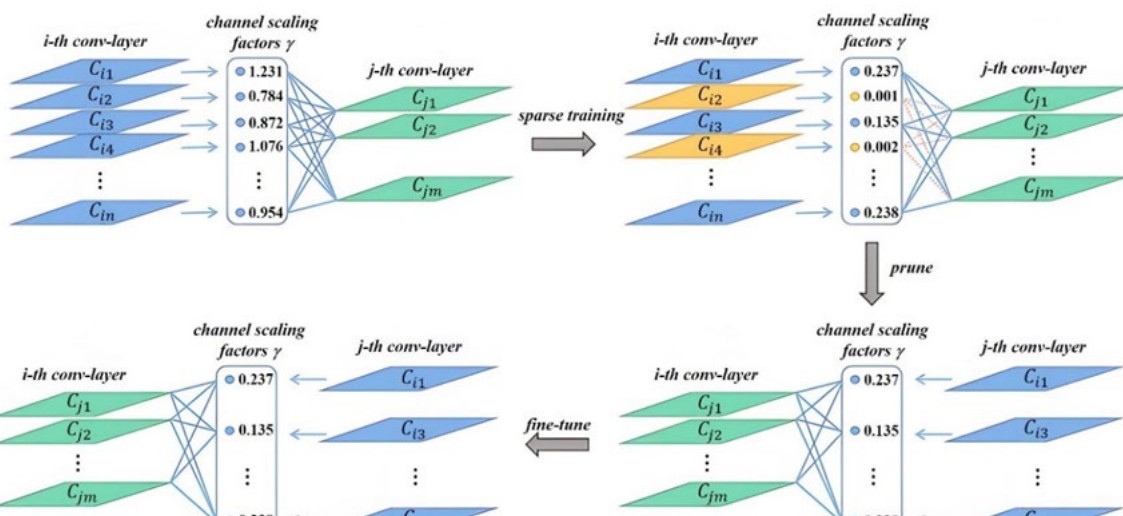

**Figure 5.** The flow of the channel pruning algorithm (The entire pruning process is divided into three steps: sparse training, pruning channel, and achieving accuracy recovery through fine-tuning).

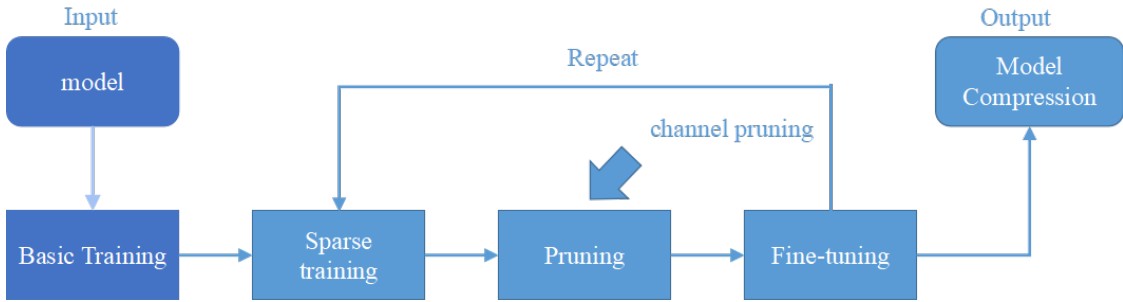

**Figure 6.** Overall flow chart of model pruning.

## 4. Experimental Analysis and Analysis

### 4.1. Experimental Environment, as well as the Dataset

     This experiment requires outstanding hardware configuration and a GPU acceleration operation. The construction, result test and training of the algorithm are all carried out under PyTorch architecture, and the CUDA parallel computing structure is adopted. Then,

the CU-DNN acceleration library is integrated, and the computer computing power is further increased under PyTorch architecture. The operating environment required for the experiment is shown in Table 1. The dataset used in this experiment is the HHD [43] safety-helmet-wearing detection dataset, which is an open-source data set proposed in 2022, including 7076 images, which are divided into training set and test set in a 9:1 ratio. The positive samples in this data set are mainly construction workers wearing helmets on the construction site; most of the negative samples are construction workers without helmets on the construction site and a few construction workers wearing other interference hats on the construction site, which has the characteristics of real construction scenes.

**Table 1.** Experiment Operating Environment (List of equipment and versions required for the experiment).

| Class | Itemize | Edition |
|---|---|---|
| Hardware configuration | system | Ubuntu 18.04 |
| | Graphics card | GeForce RTX 2080 Ti |
| Software configuration | CPU | AMD Ryzen 7 3800X 8-Core |
| | Python Version | 3.8 |
| | Deep learning framework | Pytorch |
| | CUDA | 10.0 |

*4.2. Experimental Results and Analysis*

4.2.1. MCA-YOLOV5s Fusion Analysis

In order to further fairly verify that the application of the MCA attention module to the backbone network location of the YOLOV5 model can extract more abundant features, this paper integrates the MCA attention module to different positions of the network model and studies the detection results. According to the YOLOv5s network model structure, the MCA attention module is separately fused in three regions of YOLOv5s backbone network, neck and prediction module. Since MCA modules are enhanced in important channels and spatial locations, this paper integrates the MCA attention module into each of the above three parts to produce three new network models based on the YOLOv5s algorithm: MCA-YOLOv5-BackBone, MCA-YOLOv5-Neck and MCA-YOLOv5-Prediction. Figure 7 shows the specific location of the MCA attention module fusion network.

In Figure 7a, in the backbone network of YOLOv5s, the MCA attention module is fused to CSP1_3 (i.e., feature fusion); in Figure 7b, the MCA attention module is fused to the neck Concat layer of YOLOv5s and in Figure 7c, one MCA attention module is fused before the convolution of each prediction module of YOLOv5s.

The results of the experimental comparison between the fused MCA attention modules in three different positions and the unfused MCA attention modules in three different locations are specifically shown in Table 2. Secondly, the outputs of the same channels of the three fusion design networks are compared. Subsequently, in order to observe the detection capability of the network integrating the MCA attention module relatively clearly on the targets of various sizes, in this paper the instances are classified into the following three proportional types according to the target size: small target (target area $32^2$), medium target ($32^2$ < Target area of $96^2$) and large target (target area > $96^2$). Among them, the IoU threshold of 0.5 is used for both the large and medium targets and the mAP evaluation index. Precision (P) is the correct ratio of the true values in the data predicted to be correct, and Recall (R) presents how much of the data with correct true values can be predicted correctly. Means Average Precision (mAP) refers to calculating the area under the PR curve enclosed by Precision and Recall and then averaging the categories to evaluate the detection accuracy of multiple categories.

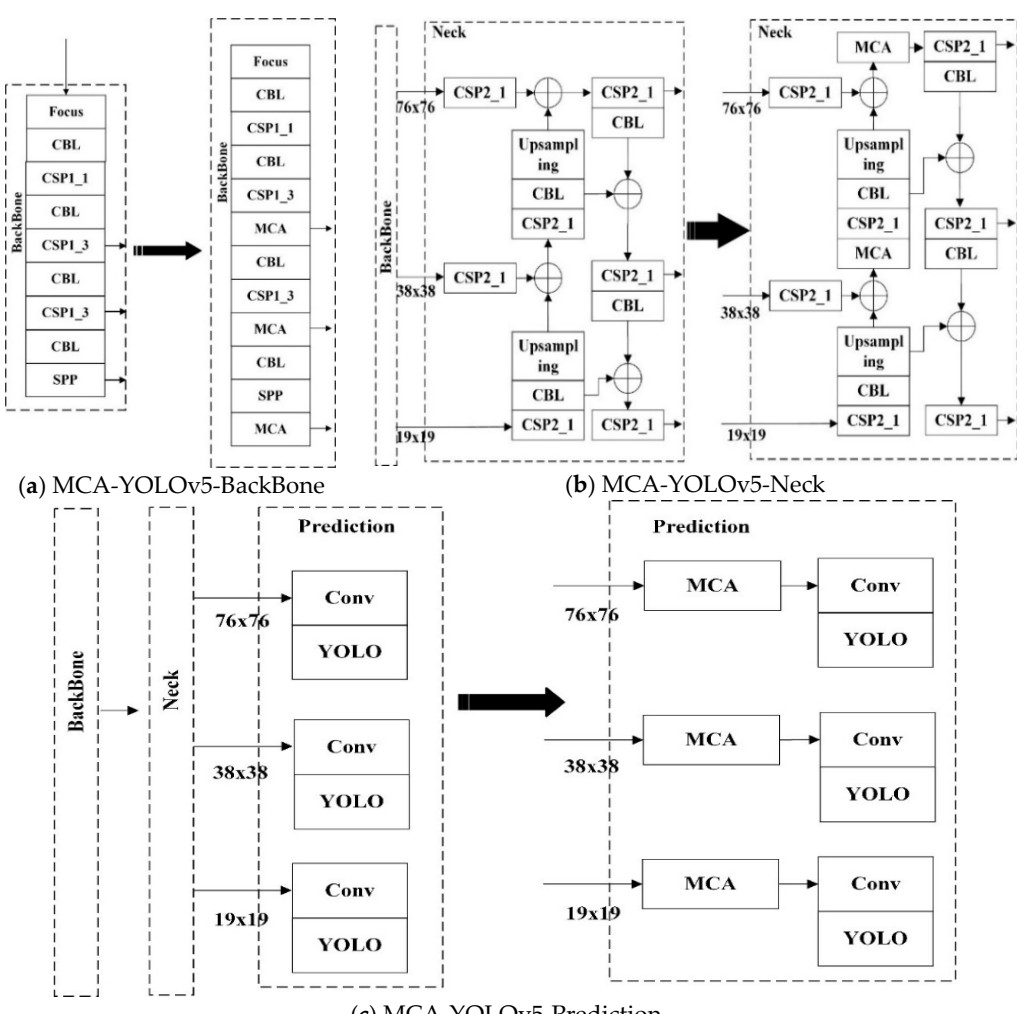

(**a**) MCA-YOLOv5-BackBone  (**b**) MCA-YOLOv5-Neck

(**c**) MCA-YOLOv5-Prediction

**Figure 7.** The YOLOv5s model of the three fused MCA modules (This is a visual representation of the MCA module placed in three different positions on the YOLOV5 model. (**a**) put the MCA module in the backbone network; (**b**) means put the MCA in the neck network and the MCA in (**c**) is in the prediction part).

**Table 2.** Comparison of the MCA module fusion results (The results of large, medium and small object detection when the MCA module located at backbone network, neck and prediction parts in the YOLOv5s.).

| Network Model | Small Object | Medium Object | Large Object | P/% | R/% | mAP/% |
|---|---|---|---|---|---|---|
| YOLOv5 | 83.0 | 97.9 | 99.3 | 76.4 | 92.5 | 92.7 |
| MCA-YOLOv5-BackBone | 90.4 | 98.6 | 99.6 | 82.2 | 95.4 | 96.0 |
| MCA-YOLOv5-Neck | 78.3 | 96.4 | 99.1 | 70.9 | 93.7 | 91.6 |
| MCA-YOLOv5-Prediction | 82.7 | 97.1 | 99.2 | 72.5 | 92.8 | 92.4 |

We analyzed the different effects of different positions of the MCA embedded model in the experimental results. Compared with the deep network, the backbone network has less semantic information but still contains the texture information and contour information that are easily ignored in the middle and low layers of the target. Moreover, the shallow information of the backbone network contains rich location information. The fusion of the MCA module in the backbone network can better fuse the easily ignored spatial features, rich location information and channel features of small objects in the feature map, thereby enhancing the feature information. By contrast, in the deeper neck of the network and

the prediction module, the feature maps have richer semantic features, smaller-sized feature maps and large receptive fields, but it is difficult for the MCA attention module to distinguish important spatial and channel features.

### 4.2.2. Sparse Training Process

The MCA-YOLOv5 algorithm was first sparsely trained; then, all the scaling factors in the BN layer were acquired simultaneously and used as a measure of the network and convolutional channels pruning. In the sparse training, if the sparsity rate is too large, the thinning rate of the model will be accelerated, but the average accuracy of the model detection method decreases significantly; on the contrary, if the sparse process is slow, the average accuracy of the model after sparse training decreases less. The sparsity algorithm is similar to a game process, often pursuing a high compression ratio but, at the same time, hoping that the sparse model can restore the original mean average accuracy and set a different sparsity rate; the results are often different, so finding the appropriate sparsity usually requires a large time cost. After repeated experiments and testing, this chapter finally set the sparse rate to 0.005. During sparse training, the change of mAP of the model on the test set during the test training is shown in Figure 8, and the change of BN layer scale factor during normal training and sparse training is shown in Figures 9 and 10, respectively.

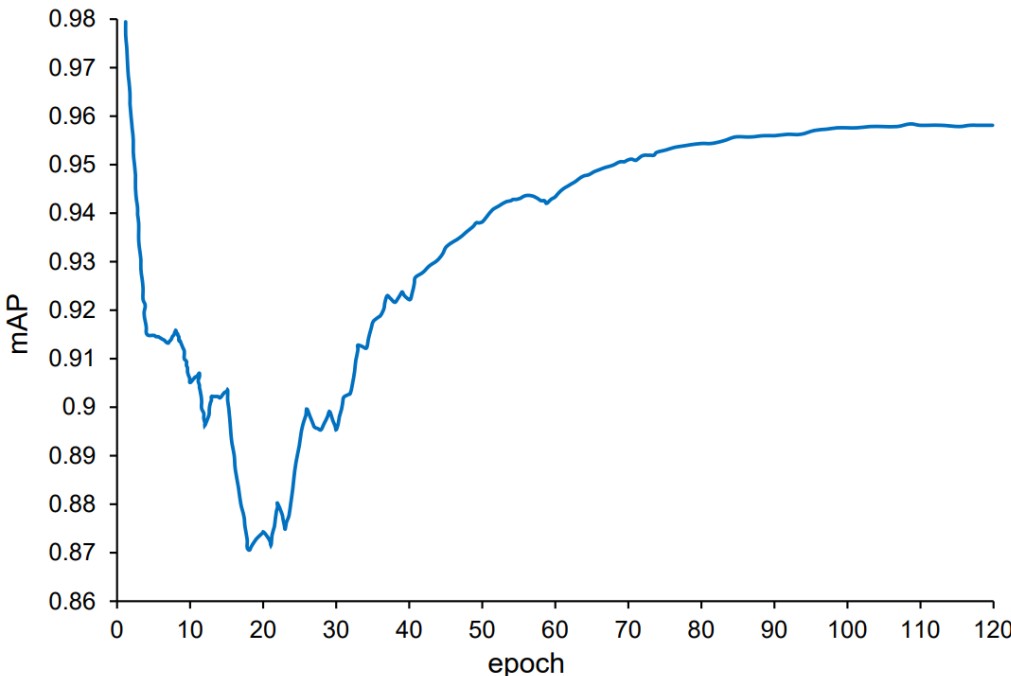

**Figure 8.** Plot of model mAP changes during sparse training.

According to Figure 9, with the increasing training rounds during the model's normal training period, the BN layer's scaling factor converges from a scattered irregular distribution to a Gaussian distribution centered on 1. Figures 9 and 10 show that the mean average accuracy of the network continuously decreases, and the values are continuously compressed in the first 20 rounds after the start of sparse training. Until the 20th epoch of sparse training, most of the values were compressed to near zero, at which point the mAP of the test set model drops to around 87%. After the 20th epoch, the mAP value of the network on the validation set gradually begins to slowly recover. Until the sparse training reaches the 110th epoch, the mAP converges to 95.8%, and the sparse training is completed when the compressed model mean average accuracy returns to the normal level. In sparse training, the weight distribution changes and mAP decreases, but after retraining, mAP will show an upward trend. After the model is fine-tuned, mAP will reach the optimal value. The scaling factor values are close to 0, meaning that the neural network

convolutional channel has low importance to the entire model, so the channel can be cut off directly during pruning and does not have much impact on the mean average accuracy of the model.

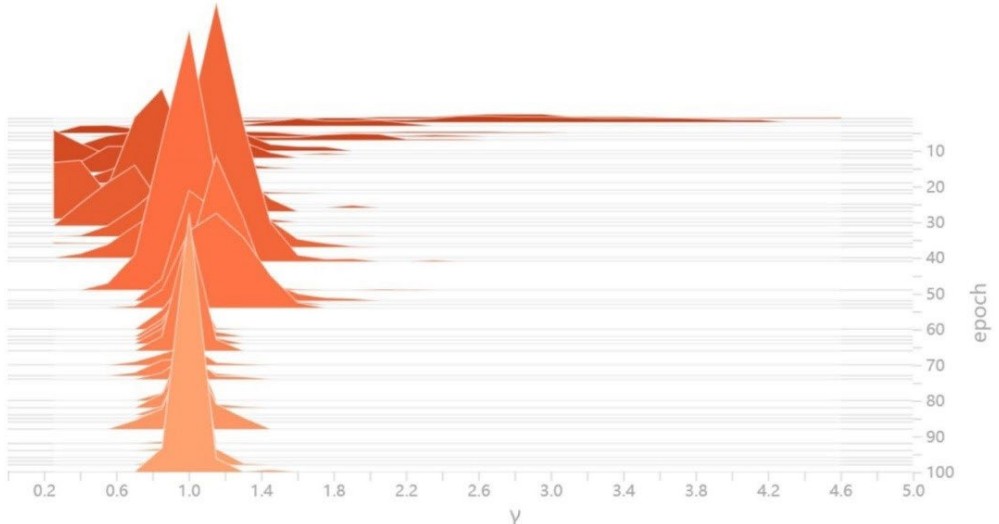

**Figure 9.** Distribution of BN layer scaling factor $\gamma$ before sparse training.

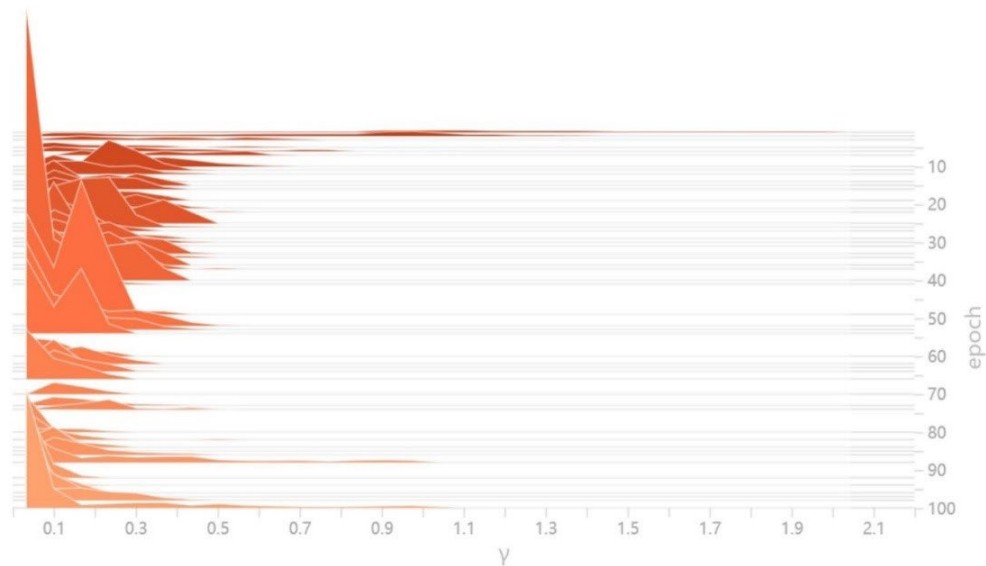

**Figure 10.** BN layer scale factor $\gamma$ distribution after sparse training.

### 4.2.3. Analysis of the Model Pruning Results

Model Pruning Comparison

After determining the scaling factor for sparse training $\gamma = 0.005$, to verify the effectiveness of model pruning, this chapter prunes the sparse trained model at rates of 0.5, 0.6, 0.7 and 0.8. Table 3 shows the parameter comparison of the detection model under different channel pruning rates and finally selects the best pruning ratio by the parameter performance.

**Table 3.** Parameter comparison of the detection models under different channel pruning rates (This table mainly compared the model running speed, size and mAP under different pruning rates).

| Cut Branch Ratio | mAP/% | Parameters/10⁶ | Reference Time/ms | Model Size/MB |
|---|---|---|---|---|
| base (0.00) | 96.0 | 7.30 | 37 | 15.2 |
| 0.80 | 89.5 | 0.93 | 17.2 | 2.1 |
| 0.70 | 93.2 | 1.67 | 18.5 | 3.6 |
| 0.60 | 93.9 | 2.39 | 19.3 | 5.0 |
| 0.50 | 94.6 | 2.96 | 20.3 | 6.1 |

Model parameters with different pruning rates were compared. According to Table 3, the number of parameters, algorithm volume and inference time of the four algorithms are reduced at different channel pruning rates. After performing fine-tuning training of the four pruning algorithms, mAP was recovered to 95.1%, 95.3%, 95.8% and 96.1%, respectively. Because the mean average accuracy after fine-tuning is not very different, and there are significant differences in the number of parameters, algorithm volume and inference time, the 80% channel pruning rate is finally selected to obtain a better model compression effect with a small loss of mean average accuracy.

Model Comparison Experiment

To verify the reliability of the algorithms proposed in this chapter, the current more popular target detection algorithms were compared with the algorithms presented in this paper, including Faster R-CNN, SSD, YOLOv3, YOLOv3 + SPP, and YOLOv5s, and the results are shown in Table 4.

**Table 4.** Comparison of the test results for the different models (Our algorithm compared with other algorithms in terms of computational speed, model size, and mAP).

| Detection Model | Hat | Person | mAP/% | FLOPS | Parameters/10⁶ | Reference Time/ms | Model Size/MB |
|---|---|---|---|---|---|---|---|
| Faster R-CNN | 80.8 | 42.2 | 61.5 | 181.12 | 186 | 291 | 182.1 |
| SSD | 78.8 | 68.2 | 73.5 | 31.75 | 23.75 | 126 | 188 |
| YOLOv3 | 89.12 | 80.7 | 84.9 | 65.86 | 61.9 | 69 | 236 |
| YOLOv3+SPP | 90.5 | 86.3 | 88.41 | 141.45 | 63.0 | 70 | 237.4 |
| YOLOv5s | 93.3 | 91.7 | 92.7 | 17.0 | 7.26 | 36 | 14.8 |
| MCA-YOLOv5 | 96.7 | 95.2 | 96.0 | 21.75 | 7.30 | 37 | 15.2 |
| MCA-YOLOv5-Light | 95.7 | 94.6 | 95.1 | 2.74 | 0.93 | 17.2 | 2.1 |

The partial detection results of the MCA-YOLOv5-Light safety helmet detection algorithm on the test sample set are shown in Figure 11. The red rectangular boxes represent the helmet-wearing situation detected by the model, and the blue boxes represent the head without a helmet. As can be seen from Figure 11, our detector can still more accurately detect whether a worker is wearing a helmet in a crowded situation.

Compared with the data in Table 4, it is clear that the MCA-YOLOv5-Light helmet-wearing detection algorithm is more effective than several popular target detection algorithms at present. Secondly, although the average mean detection accuracy of the MCA-YOLOv5-Light model was slightly lower compared with the MCA-YOLOv5 model, its model size, number of parameters and inference time were 7.2 times, 7.8 times and 2.1 times that of the MCA-YOLOv5 model, respectively. By comparing the above main evaluation indexes, the overall performance of the helmet-wearing detection algorithm presented in this paper is effective and feasible, especially in terms of the detection speed, which can meet the needs of real-time detection to a greater extent.

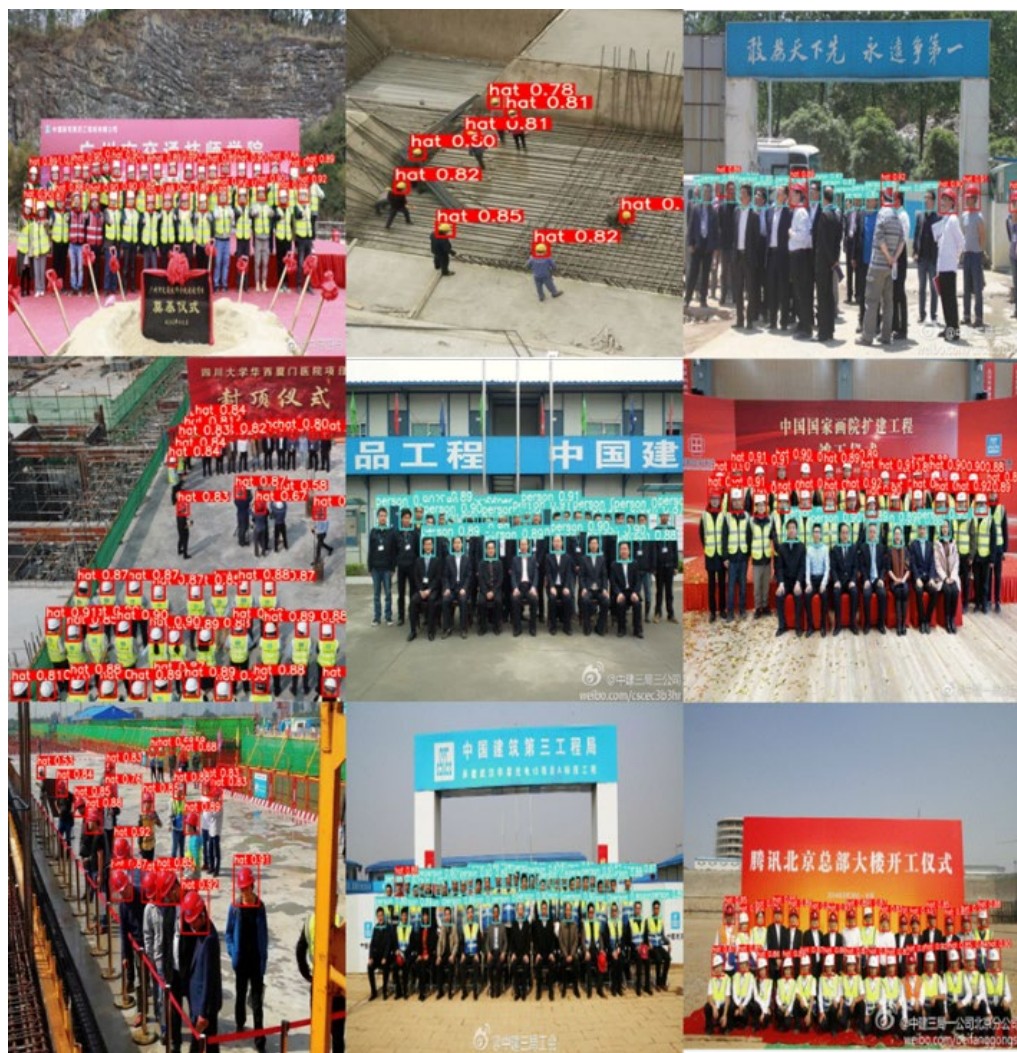

**Figure 11.** Test Results. (The figure is a real construction scene map, Chinese represents the project name. The first picture on the left in the first row is the groundbreaking ceremony of the Guangzhou Traffic Technician College, and the first picture on the right has a white title: Dare to be the first in the world, always strive to be the first. The second row from left to right are the capping ceremony of West China Xiamen Hospital of Sichuan University, China Construction, and the completion ceremony of the expansion project of China National Academy of Painting. The first picture and the second picture from the right in the third row represent the groundbreaking ceremony of the Tencent Beijing Headquarters Building and the China Construction Third Engineering Bureau.).

## 5. Conclusions

In this paper, the MCA-YOLOV5 model is proposed, embedding the MCA module in YOLOV5 to obtain more abundant feature map information. Secondly, the main strategies of sparse training and channel pruning used in this chapter are introduced. Then, the proposed strategy is adopted to implement the pruning operation on the MCA-YOLOv5 helmet-wearing detection algorithm and finally tested on the test set, and it achieved good results. Although the average accuracy of the mean detection of the MCA-YOLOv5-Light helmet wearing detection algorithm is reduced, the number of model parameters, model reasoning time, and model size of the algorithm are significantly improved compared with the MCA-YOLOv5 helmet-wearing detection algorithm, which can be used for real-time detection under the construction site.

## 6. Discussion

In this paper, we completed the task of detecting helmet wearing in the construction site scene. Although we achieved high-accuracy detection with a small model, there are some problems. The data set collected in this paper comes from web crawling, and has three sizes of helmet object: large, medium and small. If applied to real construction scenarios, it is possible that smaller helmet detection tasks are more common and important. In the next work, we will further improve the small object detection scenario and extend the algorithm to more application levels.

**Author Contributions:** Investigation, Y.W.; Validation, C.S. and J.Z.; Writing—original draft, P.Q.; Writing—review & editing, X.W. and S.Z.; Data curation, P.F. and Z.T. All authors have read and agreed to the published version of the manuscript.

**Funding:** This work is supported by the Scientific and Technological Progress and Innovation Program of the Transportation Department of Hunan Province (201927), National Defense Basic Scientific Research Project of State Administration of Science, Technology and Industry for National Defence (WDZC20205500119), the Natural Science Foundation of Hunan Province (2021JJ30456), the Open Research Project of the State Key Laboratory of Industrial Control Technology (No. ICT2022B60).

**Institutional Review Board Statement:** Not applicable.

**Informed Consent Statement:** Not applicable.

**Data Availability Statement:** https://github.com/Shiso-Q/Safety-helmet-detection/ (accessed on 1 December 2021).

**Conflicts of Interest:** The authors declare no conflict of interest.

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
