# Peer review of "MCA-YOLOV5-Light: A Faster, Stronger and Lighter Algorithm for Helmet-Wearing Detection"

_applsci, doi:10.3390/app12199697_

Round 1

Reviewer 1 Report

Authors modified YOLO version 5 architecture using pruning to detect helmets in vision data useful for construction safety assessment. I think the application and the challenge are interesting. However, there is room for improvement. Suggestions before considering for publication:

1- MCA ? what is stand for? you need to explain this in detail for general readers. 

2- The model structure is so convoluted and blood. Designed another figure for your model structure, with better quality. The input image is not a construction scene I think. This is true for other figures with blurred details. Just by looking at Figure 3, I can't understand what's going on. MCS and DCT are explained in one line. Elaborate on this section more. Why do you use FCT? I think DCT adds more computational cost to the model, am I right? Should we calculate DCT for each frame?

3- Table 3 is unnecessary, you can explain that in one sentence 

4- Honestly, I couldn't understand how MCA works. You used MCA instead of GAP. GAP is to pool everything into one vector, and MCA decreases extra information such as PCA. I like the idea but explain section 3.1.2. with more details. and with a better figure. 

5- with lots of modules in your architecture, have you done any ablation study for parts of your model?

6- Figure 8 has a separate caption. Also, why did mAP go down and returned?

7- section 4.2.3 has two subsections. so then it can be 4.2.3.1 or with no title in subsections. 

8- Figure 11 is good, make it bigger and explain scenarios. 

9- There are various ideas for attention in the literature, please cite all of them if possible. It can be a table of various attention ideas for object detection and segmentation, for instance: Mozaffari, M. Hamed, and Won-Sook Lee. "Semantic Segmentation with Peripheral Vision." International Symposium on Visual Computing. Springer, Cham, 2020.

In general, the idea and technique of the article are good, but the presentation is poor. The first pages are better than the last pages. I suggest reading the article as a general audience, and then revise it in simple and more to-the-point figures and tables. 

Reviewer 2 Report

By employing the Multispectral Channel Attention (MCA) attention mechanism and a channel pruning strategy, the authors have proposed a light-weighted approach for helmet-wearing detection based on YOLOv5. While the authors’ concern for TBI by the lack of helmet is very well-founded, the authors’ manuscript is not ready for publication due to limited scientific contribution. Specifically, the authors appear to have only provided a technical report for an adoption of MCA and a channel pruning strategy on YOLOv5, instead of a general streamlining solution. At least the following must be addressed:

-          The manuscript is specific for detection of small safety helmets, which is only a small subset of the Small Object Detection task. The authors’ approach ought to be used on larger datasets of small object detection tasks.

-          In the literature review, the authors must focus on a comparative discussion with other SOM object detection models to verify the novelty of their approach.

-          Regarding novelty, they should address what is special about the authors’ adoption of MCA compared to other attention mechanisms, and what is special about the authors’ channel pruning strategy. Apparently, MCA and channel pruning are well-proposed approaches in the literature.

-          Regarding the dataset, they should provide more information regarding the dataset. This includes citation of the HHD safety helmet-wearing detection dataset, its properties

-          Figure 3 is too difficult to understand regarding the adoption of MCA. Please also describe in detail the discrete cosine transformation DCT function for easy follow up with the authors’ formulation. Furthermore, the authors ought to provide generalization for where to apply MCA for the SOM problem, e.g. why does MCA -Backbone work, but MCA-Neck is limited in representation?

-          Instead of just using an existing channel-pruning approach, what is the authors’ idea, intuition, and proof of generalization for the generalization algorithm?

-          Citation is missing for “Moreover, the sparse training of the network layer is fully effective only when the depth of the network model exceeds 50 layers.” at line 278 of page 7.

-          Verification is missing for “That is, while ensuring the detection rate, the pruning algorithm should be able to receive higher definition images for detection, which is also conducive to the application of mobile scenarios.” at lines 268-270, page 7. These must be verified via experimentation with multiple resolutions, and even on mobile scenario.

Lastly, the authors’ model comparison experiment must include YOLOv5 and other small object detectors for fairness.

Reviewer 3 Report

Interesting work. I do have a few comments, though.

Presentation and communication style should be improved.

1) basic abbreviations such as mAP, which most researchers are familiar with, but it is better to explain

2) Table 2 contains quantities such as P, R, and mAP, but these metrics are explained in the text.

3) Figure 8: Review the caption

4) How are small, medium, and large target areas selected? Also, why is the IoU threshold set to 0.5? It indicates that you are satisfied with only a fifty percent overlap between the predicted and actual regions.

5) Normally, the train-to-test data ratio is 7:3 or 8:2, but why did the authors choose 9:1?

6) What is the Cut branch ratio in Table 3?

7) The proposed method claimed to lower hardware costs. However, it does not appear anywhere in the text to my knowledge.

Round 2

Reviewer 1 Report

The authors revised the article, and it is improved significantly. The only point that I want to mention here is that the quality of images and tables is so poor. Revised captions with more details and improved the quality. Algorithms are a little convoluted, and it requires more simplification and adding more explanation to the caption.  

Reviewer 2 Report

It appears that the authors have addressed only a few of my previous concerns. Specifically, instead of proposing a general streamlining solution with clear comparisons against existing works in  the subfield, the authors’ simply adopts MCA and a channel pruning strategy on YOLOv5 and compare it with a generality list of object detection models. At least the following must be addressed:

-          The manuscript is specific for detection of small safety helmets, which is only a small subset of the Small Object Detection task. Can the authors’ approach ought be used on other datasets of small object detection tasks? Please make the corresponding discussions.

-          In the literature review, why have the authors not provided a discussion with other small-object-detection models to highlight their approach? Please make sure to discuss similar works in this subfield.

-          Regarding the dataset, no cited information regarding the open-source HHD dataset has been provided. Please make sure to cite it.

-          Please revise the indentation of algorithm 1.

-          Verification is missing for “That is, while ensuring the detection rate, the pruning algorithm should be able to receive higher definition images for detection, which is also conducive to the application of mobile scenarios.” at lines 268-270, page 7. These must be verified by the required FLOPS and on experimental results with mobile devices.

-          Lastly, the authors’ model comparison experiment must include YOLOv5 and other small object detectors for fairness.

Example works on small-object-detection works:

[1] Y. Lee, J. -w. Hwang, S. Lee, Y. Bae and J. Park, "An Energy and GPU-Computation Efficient Backbone Network for Real-Time Object Detection," 2019 IEEE/CVF Conference on Computer Vision and Pattern Recognition Workshops (CVPRW), 2019, pp. 752-760, doi: 10.1109/CVPRW.2019.00103.

[2] Stojnić V, Risojević V, Muštra M, Jovanović V, Filipi J, Kezić N, Babić Z. A Method for Detection of Small Moving Objects in UAV Videos. Remote Sensing. 2021; 13(4):653. https://doi.org/10.3390/rs13040653

[3] Guimei Cao, Xuemei Xie, Wenzhe Yang, Quan Liao, Guangming Shi, and Jinjian Wu "Feature-fused SSD: fast detection for small objects", Proc. SPIE 10615, Ninth International Conference on Graphic and Image Processing (ICGIP 2017), 106151E (10 April 2018); https://doi.org/10.1117/12.2304811

Reviewer 3 Report

Comments are well addressed

Author Response

Thanks very much for taking your time to review this manuscript. I

really appreciate all your affirmation for ours work! Thank you very

much and have a nice day!